# Vancomycin resistant *Enterococci* and its associated factors among HIV infected patients on anti-retroviral therapy in Ethiopia

**Belayneh Regasa Dadi** *, **Zerihun Solomon, Mheret Tesfaye**

Department of Medical Microbiology, Arba Minch University, Arba Minch, Ethiopia

* belayjanimen@gmail.com

**Data Availability Statement:** Ethical restrictions have been imposed on sharing the data underlying this study. Qualified, interested researchers may submit queries related to data access to the corresponding author (belayjanimen@gmail.com)

## Abstract

### Background

The emergence of vancomycin resistant Enterococci (VRE) has alarmed the global community due to its tendency for colonization of the gastrointestinal tract. Human Immunodeficiency Virus (HIV) patients are colonized by vancomycin resistant Enterococci than other groups. The aim of this study was to determine the incidence of vancomycin resistant Enterococci and its associated factors among HIV infected patients on Anti-Retroviral Therapy (ART).

### Methods

Institution based cross sectional study was conducted among HIV infected patients on ART at from June 1 to August 30, 2020. Socio-demographic and clinical data were collected by pre-tested structured questionnaire. Stool sample was collected and processed by standard microbiological techniques. Kirby Bauer Disc diffusion method was used to perform antimicrobial susceptibility testing. Data were entered by Epi data version 4.6.0.2 and analyzed by SPSS version 25. Bivariable and multivariable logistic regression model was used to analyze the association between dependent and independent variables. P-values in the multivariable analysis, adjusted odds ratio (AOR) and 95% confidence interval (CI) were used to determine the strength of association. P-value ≤0.05 was considered as significant.

### Results

*Enterococci* spp was isolated on 123/200 (61.50%) patients. Among these isolates, the incidence of vancomycin resistant Enterococci was 11.4% [95% CI: (6.0–17.0)]. Antimicrobial susceptibility patterns against *Enterococci* showed highest rate of resistance to ampicillin (69.9%). Multidrug resistances were observed in 49.59% of *Enterococci* isolates. Study participants who had prior antibioticexposurer more than two weeks [AOR = 7.35; 95% CI: (1.21 44.64)] and hospitalization for the last six months [AOR = 5.68; 95% CI: (1.09 29.74)] were significantly associated with vancomycin resistant *Enterococci*.

or the authors' Institutional Review Board (contact: Zerihun Zerdo; email: zedozerihun@gmail.com).

**Funding:** The authors received no specific funding for this work.

**Competing interests:** The authors have declared that no competing interests exist.

**Abbreviations:** ART, Anti-Retroviral Therapy; AOR, Adjusted odd ratio; BEAA, Bile Esculin Azide agar; CDC, Center for Disease Control; CI, Confidence Interval; CLSI, Clinical Laboratory Standards Institute guidelines; CSA, Central statistical agency; HAI, Hospital Acquired Infections; HIV, Human Immunodeficiency Virus; MDR, Multi Drug Resistant; SOP, Standard Operating Procedures; VRE, Vancomycin Resistance *Enterococcus*; WHO, World Health Organization.

## Conclusions

In our study high incidence of vancomycin resistant *Enterococci* was found. Previous exposure to antibiotics for more than two weeks and hospitalization for more than six months were significantly associated with vancomycin resistant *Enterococci*. The isolated *Enterococci* had variable degrees of resistance to commonly prescribed antibiotics. Therefore, periodic surveillance on antimicrobial resistance pattern, adhering to rational use of antibiotics and implementing infection prevention protocols may reduce colonization by VRE.

## Introduction

A major problem with the *Enterococci* is they are very resistant to antibiotics and have ability to survive in harsh environments in community and persist in hospital settings [1]. Because of this, they become important in health facility based settings [2]. According to World Health Organization (WHO) report in 2017, vancomycin resistant Enterococci (VRE) is one of the most resistant bacteria in their "Global Priority list of antibiotic-resistant bacteria" [3]. In the same manner, the Center for Disease Control and Prevention (CDC) has classified *Enterococci* among bacteria with a threat level of serious [4]. Currently, VRE are the cause of one-third and one fifth of all health care associated infections in the United States and in some European countries respectively [5].

VRE are known in causing different enterococcal infections such as infective endocarditis, bacteremia, urinary tract infection, intra-abdominal and pelvic infections, surgical wound infections, and very rarely Central nervous system (CNS) infection. Many of these infections originate from intestinal flora of colonized individuals. VRE have different selection pressures for proliferation and rapid expansion of its resistant populations. Furthermore, few options are left for management of diseases caused by VRE, and hence causing increased mortality on infected individuals. VRE now represent approximately one third of *Enterococcus* isolates [6–17].

Asymptomatic VRE gut colonization precedes infection with susceptible hosts. Such susceptible hosts are patients who are exposed to multiple and prolonged courses of antimicrobial agents like Human Immunodeficiency virus (HIV) infected individuals, severely ill, hospitalized for long lengths of stay (LOS), living in a long-term care facility, located in close proximity to another colonized or infected patient, or hospitalized in a room previously occupied by a patient colonized with VRE. Colonization is often obtained by vulnerable hosts in an environment with increased rate of patient colonization with VRE [2, 18–26]. The prevalence of VRE was reported in Europe, Asia, Australia, South America and some African countries [1, 5, 24, 26]. However, there is no sufficient data available on the prevalence and risk factors of VRE in developing countries like Ethiopia. Therefore, this study was conducted with the aim of determining the prevalence of vancomycin resistant enterococci and its associated factors among HIV infected patients on Anti-Retroviral Therapy (ART).

## Materials and methods

### Study design, period and setting

The study was conducted in Arba Minch General Hospital (Arba Minch, Ethiopia) from June 1 to August 30, 2020. The hospital has ART clinic where it provides different services for HIV-infected patients. The total population of Arba Minch town from 2007 central statistical agency

(CSA) census report was 74,879, of whom 39,208 were men and 35,671 women [27]. The inclusion criteria was all HIV infected patients on ART during the study period. Exclusion criteria were HIV infected patients who don't have Parent or guardian assent if they are <18 years old and those HIV infected patients who were critically ill and unable to respond.

## Sampling technique

Systematic random sampling technique was used to select the study participants. Based on the 2020 three months (June-August, 2020) data obtained during COVID-19 pandemic from Arba Minch General Hospital ART clinic, N = 366 (N = total number of study participants) HIV infected patients has visited the ART clinic. Assuming the same number of HIV- infected patients for the study period (June-August, 2020) and taking sample size of 200, the $k^{th}$ value (k = N/n = 366/2002) is found to be 2 [28]. The first one is selected with lottery method from $1^{st}$ and $2^{nd}$ patients and found to be $2^{nd}$ patient. Then every $2^{nd}$ patient was included.

Dependent variable was VRE gut colonization and independent variables were: Age, Sex, CD4 count, Level of hemoglobin, Previous antibiotic treatment for >2 weeks, Current visiting status (Inpatient or Outpatient), History of hospitalization in the last six months, Urinary catheterization, Co-morbid conditions (diabetes and renal failure).

## Data collection and laboratory processing

A pretested well designed structured questionnaire was used to collect data from the study participants. The questionnaire was designed by reviewing different literatures and prepared in English and Amharic languages.

Upon the arrival of each study participant at ART clinic, written assent/consent was obtained. Socio-demographic and clinical data were collected by two nurses using a pretested well designed structured questionnaire through face-to-face interview. Moreover, recent CD4 count and hemoglobin level of respondents have been taken from ART clinic logbook.

## Isolation and identification of *Enterococci*

Patients were instructed to collect about 2gm of the faecal specimen in a sterile wide-mouth screw capped container labeled with the unique sample number, date, and time of collection. The collected stool specimens were transported to Arba Minch University, College of Medicine and Health Sciences, Microbiology and Parasitology laboratory.

Immediately after delivery, the transported stool specimens was streaked on Bile Esculin Azide agar (BEAA) (Park Scientific Limited, 24 Low Farm Place, Moulton Park, Northampton, NN3 6HY) and incubated for 24 hours at 37˚C. Plates were observed for appearance of characteristic growth with brown-black colored colonies in the medium and dark halo centers. Typical characteristic colonies were selected randomly for characterization and presumptive identification of *Enterococci* by Gram stains, Catalase test, Salt tolerance test and Heat tolerance test [29].

The antimicrobial susceptibility testing was performed using Kirby Bauer disc diffusion method according to Clinical Laboratory Standards Institute guidelines (CLSI) [30]. After a pure colony was obtained, a loop full of bacteria were taken, transferred to a tube containing 5 ml of sterile normal saline (0.85% NaCl) and mixed gently until it formed a homogenous suspension. The turbidity of the suspension was determined by comparison with 0.5 McFarland standards. A sterile swab was dipped into the suspension, and excess suspension was removed by pressing the swab against the wall of the tube. The entire surface of Muller Hinton Agar plate was uniformly flooded with suspensions and allowed to dry for about 3–5 minutes. The antimicrobial impregnated disks were placed by using sterile forceps at least 24 mm away from

each other to avoid the overlapping zone of inhibition. The disks were placed on agar plates and allowed to stand for 30 minutes to dissolve the antibiotics in the media [16]. The plates were then inverted and incubated at 37$^{o}$c for 16–18 hours and inhibition zone was measured using a ruler.

In the case of vancomycin, the plates were inverted and incubated at 37$^{o}$c, and held a full 16–18 hours for accurate detection of resistance. Zones were examined using transmitted light; the presence of a haze or any growth within the zone of inhibition indicates resistance for vancomycin. Grades of susceptibility pattern were recognized as sensitive, intermediate and resistant by comparison of zone of inhibition according to the 2018 CLSI guidelines [30].

Antimicrobial susceptibility patterns of *Enterococci* were also assessed against the following antibiotic discs: Penicillin (10 IU), Ampicillin (10 μg), Tetracycline (30μg), Doxycycline (30μg), Ciprofloxacin (5μg), Vancomycin (30 μg), Erythromycin (15μg) and Chloramphenicol (30μg). Interpretations of results were made according to CLSI (30). Antibiotics were selected based on CLSI recommendation, local availability (in health facility) and feasibility (cost and method of antimicrobial susceptibility test).

## Data quality assurance

A pretest was done on 5% (n = 10) HIV infected individuals. Quality control measures were implementing throughout the entire process of data collection and laboratory work. Training was given for data and sample collectors concerning on the research objective, data collection tools, sample collection procedures, and infection prevention protocols to be taken related with COVID-19 Pandemic. Standard Operating Procedures (SOP) were prepared and implemented strictly. All culture media were prepared following the manufacturer's instruction and sterility of the culture media was tested by incubating 5% of the batch at 35–37˚C overnight for evaluation of possible contamination. Standard control strains *E. faecalis* ATCC 29212 and *Staphylococcus aureus* ATCC 25923 were used as a positive control.

## Statistical analysis

Data were checked for its completeness, entered by Epi data version 4.6.0.2 software, and analyzed by SPSS version 25. The fitness of the model was checked by Hosmer-Lemeshow goodness of fit test. Bivariable and multivariable logistic regression model was used to analyze the association between dependent and independent variables. Those variables with P-value < 0.25 in bivariate analysis were considered as candidate for further multivariable analysis. P-values in the multivariable analysis, adjusted odds ratio (AOR) and 95% confidence interval (CI) were used to determine the strength of association. P-value <0.05 was considered as statistically significant.

## Ethics approval and consent to participate

Ethical clearance was obtained from Institutional Review Board (IRB) of Arba Minch University, College of Medicine and Health Sciences, and letter of permission to conduct the study was written to Arba Minch General Hospital. Official permission from Arba Minch General Hospital and an informed written assent/consent from each study participants (assent of those <18 age group from their family or guardian) were obtained.

All information that identifies study subjects were given code numbers and were kept confidential. The purpose of the study was clearly described to the study participants and the specimens collected from the patients were analyzed for the intended purpose only. For each confirmed resistant case, the responsible clinicians of the patient were informed for appropriate management.

## Results

### Socio-demographic characteristics of study participants

A total of 200 study participants were enrolled in this study. Among these the ratio of male to female was 1:1. The mean age and standard deviation was 38.68 ± 10.97 ranging from 18–67 years. More than three fourth of the study participants 154 (77.0%) came from urban settings (**Table 1**).

### Clinical characteristics of study participants

The clinical data showed that majority of the participants 187 (93.5%) were outpatients and 168 (84.0%) of the participants had CD4 count >350. Less than half of the respondents 83 (41.5%) had low hemoglobin level and 94 (47.0%) had history of previous antibiotic use for <2 weeks. Majority of the study participants 174 (87.0%) didn't have previous history of hospitalization in the last six months and 180 (90.0%) didn't have history of previous catheterization. Likewise, majority of the respondents 191 (95.5%) and 193 (96.5%) didn't have Diabetes Mellitus and Renal failure respectively (**Table 2**).

### Incidence of vancomycin resistant *Enterococcus*

From the total of 200 study participants, *Enterococci* species was isolated from123 (61.50%) patients. Among these isolates, 14 (11.4%) were vancomycin resistant (95% CI: 6.0–17.0%).

### Antimicrobial resistance profile of isolated *Enterococci*

Among 123 *Enterococci* isolates tested for commonly prescribed antimicrobial agents the highest rate of resistance was observed against ampicillin in which more than two thirds 86 (69.9%) of the isolates were resistant (**Fig 1**).

**Table 1. Socio-demographic characteristics of study participants attending ART clinic at Arba Minch General Hospital, Southern Ethiopia October 2020 (N = 200).**

| Variables | Frequency | Percentage (%) |
|---|---|---|
| **Sex** | | |
| Males | 100 | 50.0 |
| Females | 100 | 50.0 |
| **Age** | | |
| ≤20 | 12 | 6.0 |
| 21–30 | 30 | 15.0 |
| 31–40 | 78 | 39.0 |
| 41–50 | 47 | 23.5 |
| ≥51 | 33 | 16.5 |
| **Residence** | | |
| Rural | 46 | 23.0 |
| Urban | 154 | 77.0 |
| **Educational Status** | | |
| Unable to read and write | 17 | 8.5 |
| Able to read and write | 42 | 21.0 |
| Primary (1–8) | 62 | 31.0 |
| Secondary (9–12) | 62 | 31.0 |
| College and above | 17 | 8.5 |

**Table 2. Clinical characteristics of study participants attending ART clinic at Arba Minch General Hospital, Southern Ethiopia October 2020 (N = 200).**

| Visiting Status | Frequency | Percentage |
|---|---|---|
| Outpatient | 187 | 93.5 |
| Inpatient | 13 | 6.5 |
| **CD$_4$+ Count** | | |
| ≤350 | 32 | 16.0 |
| >350 | 168 | 84.0 |
| **Hemoglobin Level** | | |
| Low | 83 | 41.5 |
| Normal | 111 | 55.5 |
| High | 6 | 3.0 |
| **Previous Antibiotic Treatment** | | |
| Never | 76 | 38.0 |
| ≥2 weeks | 30 | 15.0 |
| <2 weeks | 94 | 47.0 |
| **History of hospitalization in the last six months** | | |
| No | 174 | 87.0 |
| Yes | 26 | 13.0 |
| **History of previous catheterization** | | |
| No | 180 | 90.0 |
| Yes | 20 | 10.0 |
| **Comorbid condition (Diabetes)** | | |
| No | 191 | 95.5 |
| Yes | 9 | 4.5 |
| **Comorbid condition (Renal failure)** | | |
| No | 193 | 96.5 |
| Yes | 7 | 3.5 |

Multidrug resistances (MDR) were observed in slightly less than half 61 (49.59%) of *Enterococci* isolates and less than one tenth 11 (8.94%) of the isolates were resistant to all antimicrobials tested. All VRE isolates were MDR (**Table 3**).

## Factors associated with vancomycin resistant *Enterococcus* gut colonization

Factors independently associated with VRE gut colonization on bivariable analysis, and fit into a multivariable logistic regression model were sex, visiting status, history of catheterization, previous antibiotics treatment, history of hospitalization in the last six months and presence of Diabetes Mellitus (**Tables 4 and 5**).

## Discussion

The hasty emergence and the increasing incidence of colonization with VRE have become challenging healthcare problems that have caused serious concern to health care providers and health authorities [2]. In the present study, the prevalence of VRE among HIV infected patients was 11.4% (95% CI: 6.0–17.0%). This prevalence rate was consistent with other studies conducted in, West Amhara, Ethiopia (7.7%) [2] Gondar, Northwest Ethiopia (7.8%) [18], where these studies were done on HIV positive patients and also comparable with studies done on HIV status patients not known in Addis Ababa, Ethiopia (6.7%) [31] and South Africa (14.5%) [21]. However, the prevalence of VRE in our study is lower than the report from

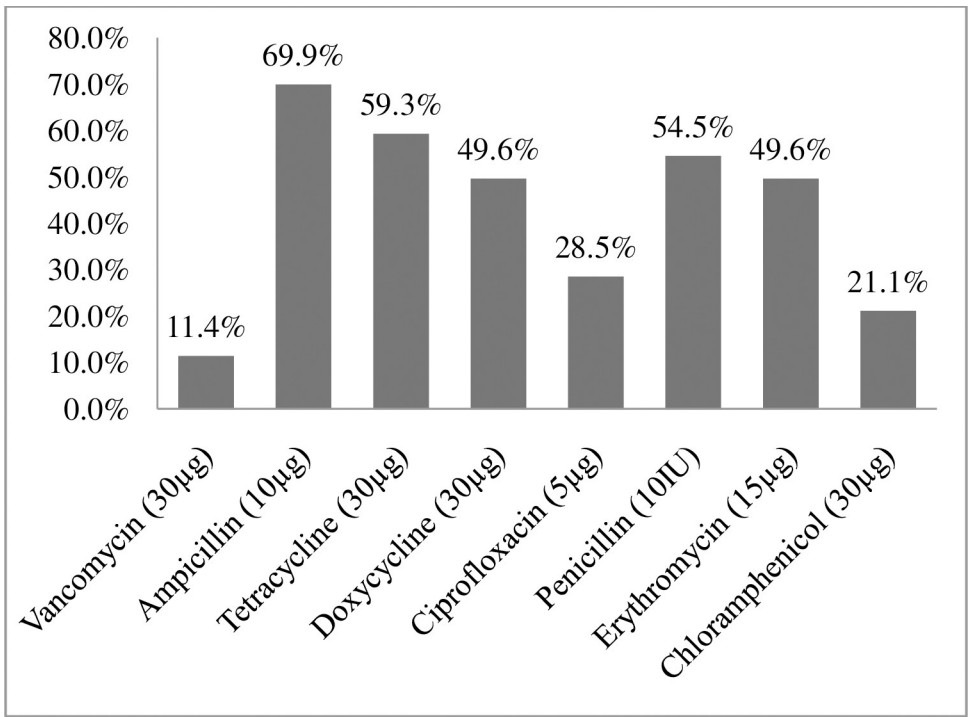

**Fig 1. Antimicrobial resistance patterns of isolated *Enterococcus* spp among clients attending ART clinic at Arba Minch General Hospital, Southern Ethiopia October 2020 (n = 123).**

Ireland (44.1%) [4], Germany (26.1%) [32] Brazil (23.4%) [1], India (19.6%) [33], Saud Arabia (17.3%) [5] and Iraq (46.4%) [17] where these studies were conducted on the population where HIV status not determined. The lower prevalence in the present study might be due to the variation in study settings in which most of the previous countries have been using oral and intravenous vancomycin massively for human diseases [34, 35], and variation in study participants where the previous studies participants were hospitalized patients and critically ill patients in ICU who were frequently exposed to different antibiotics and experienced insertion of external devices like catheter [7]. On the other hand, the prevalence in our study is higher than studies conducted in USA (4.7%) [36], Nigeria (4.07%) [37] and Ethiopia (5.9%) [16]. The gradual increase and clonal expansion in the prevalence of VRE might have contributed to this higher prevalence [32, 38].

*Enterococci* isolates showed various resistances to the tested antibiotics; namely, 69.9% to ampicillin, 54.5% to penicillin, 49.6% to erythromycin, 59.3% to tetracycline, 28.5% to

**Table 3. Profile of multidrug resistance pattern of VRE isolates among clients attending ART clinic at Arba Minch General Hospital, Southern Ethiopia October 2020 (n = 14).**

| Resistance rate | Combination of Antibiotics | No. (%) of isolates tested |
| --- | --- | --- |
| R4 | G, P, TTC, MAC | 14 (100) |
| R5 | (G, P, TTC, MAC) + F | 2 (14.28) |
| R6 | (G, P, TTC, MAC) + F+ PH | 11 (78.58) |

**Key;** G-glycopeptides (vancomycin), P-penicillins (ampicillin and/or penicillin), TTC-tetracyclines (tetracycline and/or doxycycline), MAC-macrolides (erythromycin), F-fluoroquinolones (ciprofloxacin), PH-phenicols (chloramphenicol), and R4-R6 Number of categories of antibiotics resistance from 4 to 6, respectively

**Table 4. Bivariable logistic regression analysis on factors affecting vancomycin resistant *Enterococci* among clients attending ART clinic at Arba Minch General Hospital, Southern Ethiopia October 2020.**

| Characteristics | Categories | VRE gut colonization | | COR | P-value |
|---|---|---|---|---|---|
| | | Yes, N (%) | No, N (%) | | |
| Sex | Females | 4 (6.8) | 55 (93.2) | **0.39** | **0.133**[*] |
| | Males | 10 (15.6) | 54 (84.4) | 1 | |
| Residence | Urban | 11 (11.6) | 84 (88.4) | 1.09 | 0.899 |
| | Rural | 3 (10.7) | 25 (89.3) | 1 | |
| Visiting status | Inpatient | 3 (30.0) | 7 (70.0) | **3.97** | **0.069**[*] |
| | Outpatient | 11 (9.7) | 102 (90.3) | 1 | |
| CD4 count | ≤350 | 2 (11.8) | 15 (88.2) | 1.04 | 0.957 |
| | >350 | 12 (11.3) | 94 (88.7) | 1 | |
| Previous Antibiotic Treatment | Never | 2 (4.9) | 39 (95.1) | 1 | |
| | ≥2 weeks | 8 (40.0) | 12 (60.0) | **13.00** | **0.003**[*] |
| | <2 weeks | 4 (6.5) | 58 (93.5) | 1.34 | 0.739 |
| History of hospitalization in the last six months | Yes | 6 (37.5) | 10 (62.5) | **7.43** | **0.002**[*] |
| | No | 8 (7.5) | 99 (92.5) | 1 | |
| History of previous catheterization | Yes | 4 (25.0) | 12 (75.0) | **3.23** | **0.078**[*] |
| | No | 10 (9.3) | 97 (90.7) | 1 | |
| Diabetes Mellitus | Yes | 2 (28.6) | 5 (71.4) | **3.48** | **0.163**[*] |
| | No | 12 (10.3) | 104 (89.7) | 1 | |

[*]Variables which passed bivariable logistic regression analysis at cut off value <0.25

ciprofloxacin and 21.1% to chloramphenicol. These findings are comparable with studies conducted in India 64.9% [33] and Ethiopia 66.7% [39] for ampicillin; Ethiopia 64.9% [40], 68% [39] for tetracycline; and Brazil 45.7% [1] and Ethiopia 42.7% [9] 53.3% [31] for erythromycin. However, the resistance profiles in our study is lower than previous studies in India for tested antibiotics 75.9% for penicillin, 84.5% for tetracycline, 95.5% for ciprofloxacin, 92.1% for

**Table 5. Multivariable logistic regression analysis on factors affecting vancomycin resistant *Enterococci* among clients attending ART clinic at Arba Minch General Hospital, Southern Ethiopia October 2020.**

| Characteristics | Categories | VRE isolates | | AOR (95%CI) | P-value |
|---|---|---|---|---|---|
| | | Yes, N (%) | No, N (%) | | |
| Sex | Females | 4 (6.8) | 55 (93.2) | 0.27 (0.06–1.25) | 0.093 |
| | Males | 10 (15.6) | 54 (84.4) | 1 | |
| Visiting status | Inpatient | 3 (30.0) | 7 (70.0) | 1.13 (0.15–8.43) | 0.904 |
| | Outpatient | 11 (9.7) | 102 (90.3) | 1 | |
| Previous Antibiotic Treatment | Never | 2 (4.9) | 39 (95.1) | 1 | |
| | ≥2 weeks | 8 (40.0) | 12 (60.0) | **7.35 (1.21–44.64)** | **0.030**[**] |
| | <2 weeks | 4 (6.5) | 58 (93.5) | 1.08 (0.17–6.80) | 0.936 |
| History of hospitalization in the last six months | Yes | 6 (37.5) | 10 (62.5) | **5.68 (1.09–29.74)** | **0.040**[**] |
| | No | 8 (7.5) | 99 (92.5) | 1 | |
| History of previous catheterization | Yes | 4 (25.0) | 12 (75.0) | 1.38 (0.27–7.03) | 0.701 |
| | No | 10 (9.3) | 97 (90.7) | 1 | |
| Diabetes Mellitus | Yes | 2 (28.6) | 5 (71.4) | 1.81 (0.21–15.36) | 0.588 |
| | No | 12 (10.3) | 104 (89.7) | 1 | |

[**]Significant in multivariable logistic regression at p-value <0.05

erythromycin and 42.3% for chloramphenicol [33]; Uganda 69.4% [15] and Ethiopia 77.3% [41] for tetracycline; and Iraq (85.7%) [17], Uganda (72%) [15] and Ethiopia 63.2% [41] erythromycin; These lower drug resistance patterns might be due to variations in sample size, methodology and study participants where most of the participants in the previous studies were hospitalized patients who were taking different antibiotics that might have been contributed for emergence of high rate of drug resistance.

On the other hand, the antibiotic resistance profile in our study is higher than studies conducted in Iran 41.2% [42], Brazil 0% [1], Uganda 1.4% [15] and Ethiopia 36% [41] for ampicillin; Brazil 32.6% [1] and Ethiopia 37.7% [2] for tetracycline; and Brazil 10.9% [1] for chloramphenicol. The high drug resistance patterns might be due to overuse or misuse of antibiotics and inappropriate antibiotics prescription which is very common practice in Ethiopia. Additional reasons for high drug resistance might be extensive antibiotics use of antibiotics for agricultural purpose, mutation and gene transfer among bacteria [4].

Moreover, the present study also showed that 49.6% of *Enterococci* isolates were multidrug resistant. This result is lower than the findings in Iraq (85.7%) [17] and Ethiopia 80.8% [41]. In contrary, the result is higher than the finding in Ethiopia 29.5% [16]. The discrepancy of the result might be due to variation in geographical distribution of strain, trend and frequency of antibiotic prescription, community drug usage practice.

Our study showed that HIV patients who were previously exposed to antibiotics for more than two weeks were seven times more likely to be colonized with VRE as compared with HIV patients who never exposed to antibiotics previously [AOR = 7.35; 95% CI: (1.21–44.64); P-value = 0.030]. This result is in agreement with other studies conducted in Brazil [1], South Korea [43], Egypt [44], and Ethiopia [2]. The reason might be a prior exposure to antibiotics for a prolonged duration can cause VRE colonization due to the fact that the antibiotics exert selective pressure to Enterococci and alter the competing microbial flora in the GI tract allowing VRE to predominate as evidenced by other studies [7, 34].

The present study showed that HIV patients who had history of hospitalization for the last six months were six times more likely to be colonized with VRE as compared with HIV patients who didn't have history of hospitalization for the last six months [AOR = 5.68; 95% CI: (1.09–29.74); P-value = 0.040]. The finding is consistent with previous studies done in USA [36], South Korea [43] and Ethiopia [2]. The reason might be VRE have been isolated from virtually every object within patient rooms since they are intrinsically resistant to several commonly used antibiotics in hospital and have ability to acquire resistance genes. Besides, they are ubiquitous in their presence and have high survivability on dry surfaces, thereby causing high VRE transmission rates within healthcare facilities [7, 9].

## Limitations of the study

The isolated *Enterococci* were not identified to species level; molecular characterization and MIC were not done due to resource limitation and budget constraints.

## Conclusions

In our study high incidence of vancomycin resistant Enterococci was found. Previous exposure to antibiotics for more than two weeks and previous hospitalization for more than six months were significant factors for vancomycin resistant enterococci gut colonization. The study also showed that the isolated *Enterococci* had variable degrees of resistance to commonly prescribed antibiotics.

Therefore, periodic surveillance on antimicrobial resistance pattern, adhering to rational use of antibiotics and implementing infection prevention protocols may reduce colonization by VRE.

## Supporting information

**S1 File.**
(PDF)

**S2 File.**
(DOCX)

## Acknowledgments

The authors would like to thank those who were involved in this research.

## Author Contributions

**Conceptualization:** Belayneh Regasa Dadi, Zerihun Solomon, Mheret Tesfaye.

**Data curation:** Belayneh Regasa Dadi.

**Formal analysis:** Belayneh Regasa Dadi, Zerihun Solomon.

**Funding acquisition:** Belayneh Regasa Dadi.

**Investigation:** Belayneh Regasa Dadi, Zerihun Solomon, Mheret Tesfaye.

**Methodology:** Belayneh Regasa Dadi, Zerihun Solomon, Mheret Tesfaye.

**Project administration:** Belayneh Regasa Dadi, Mheret Tesfaye.

**Resources:** Belayneh Regasa Dadi, Mheret Tesfaye.

**Software:** Belayneh Regasa Dadi, Mheret Tesfaye.

**Supervision:** Belayneh Regasa Dadi, Mheret Tesfaye.

**Validation:** Belayneh Regasa Dadi, Mheret Tesfaye.

**Visualization:** Belayneh Regasa Dadi.

**Writing – original draft:** Belayneh Regasa Dadi, Zerihun Solomon, Mheret Tesfaye.

**Writing – review & editing:** Belayneh Regasa Dadi, Zerihun Solomon, Mheret Tesfaye.

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
