## [Decision Letter · Decision Letter 0]

17 Dec 2020

PONE-D-20-34817

Vancomycin resistant enterococci and its associated factors among HIV infected patients on anti-Retro viral therapy in Ethiopia

PLOS ONE

Dear Dr. Regasa Dadi,

Thank you for submitting your manuscript to PLOS ONE. After careful consideration, we feel that it has merit but does not fully meet PLOS ONE’s publication criteria as it currently stands. Therefore, we invite you to submit a revised version of the manuscript that addresses the points raised during the review process.

Discussion section could focus on interpreting the results rater than repeat the points made in the results section

We look forward to receiving your revised manuscript.

Kind regards,

Iddya Karunasagar

Academic Editor

PLOS ONE

Journal Requirements:

7. We note you have included a table to which you do not refer in the text of your manuscript. Please ensure that you refer to Table 5 in your text; if accepted, production will need this reference to link the reader to the Table.

8. Please include your tables as part of your main manuscript and remove the individual files. Please note that supplementary tables (should remain/ be uploaded) as separate "supporting information" files

Additional Editor Comments:

The reviewers have commented on the manuscript and raised number of pertinent points that need to be addressed. Discussion section should interpret the results and discuss in the light of current knowledge rather than repeating the results. This applies to Conclusion section abstract too. Please revise the manuscript addressing all reviewer comments.

Reviewers' comments:

Reviewer's Responses to Questions

**Comments to the Author**

1. Is the manuscript technically sound, and do the data support the conclusions?

Reviewer #1: Yes

2. Has the statistical analysis been performed appropriately and rigorously? 

Reviewer #1: Yes

3. Have the authors made all data underlying the findings in their manuscript fully available?

Reviewer #1: Yes

4. Is the manuscript presented in an intelligible fashion and written in standard English?

Reviewer #1: No

5. Review Comments to the Author

Reviewer #1: 1. The language should be improved.

2. The manuscript presents extensive introduction which can be reduced to two or three paragraphs.

3. Statistical tests should be mentioned in the abstract also.

4. In methodology, some sentences can be eliminated like the section on antimicrobial susceptibility testing.

5. The name of the institute should be masked.

6. Discussion needs to be improved. Rather than stating the facts, the results should be compared and justification (or hypotheses) for any differences should be explored.

7. The disk diffusion test alone was used. Explain why MIC was not estimated and species level identification was not performed.

6. PLOS authors have the option to publish the peer review history of their article (what does this mean?). If published, this will include your full peer review and any attached files.

Reviewer #1: No

---

## [Author Response · Author response to Decision Letter 0]

18 Jan 2021

Point by point response

1. Response to editors

Comment given: Please ensure that your manuscript meets PLOS ONE's style requirements, including those for file naming. The PLOS ONE style templates can be found at https://journals.plos.org/plosone/s/file?id=wjVg/PLOSOne_formatting_sample_main_body.pdf andhttps://journals.plos.org/plosone/s/file?id=ba62/PLOSOne_formatting_sample_title_authors_affiliations.pdf

Response: Comment accepted and corrections were made following the guideline and sample template. (Check for tracking documents where changes made accordingly)

Comment given: Please include additional information regarding the survey or questionnaire used in the study and ensure that you have provided sufficient details that others could replicate the analyses. For instance, if you developed a questionnaire as part of this study and it is not under a copyright more restrictive than CC-BY, please include a copy, in both the original language and English, as Supporting Information.

Response: Comment accepted and questionnaire (both in local language and English) included as additional information.

Comment given: Thank you for stating the following financial disclosure:

a. Please clarify the sources of funding (financial or material support) for your study. List the grants or organizations that supported your study, including funding received from your institution.

d. If you did not receive any funding for this study, please state: “The authors received no specific funding for this work.”

Response: Comment accepted financial disclosure mentioned in the manuscript “The funders had no role in study design, data collection and analysis, decision to publish, or preparation of the manuscript. The authors received no specific funding for this work” (see page 17 paragraph 1)

Comment given: We note that you have indicated that data from this study are available upon request. PLOS only allows data to be available upon request if there are legal or ethical restrictions on sharing data publicly. In your revised cover letter, please address the following prompts: a) If there are ethical or legal restrictions on sharing a de-identified data set, please explain them in detail (e.g., data contain potentially identifying or sensitive patient information) and who has imposed them (e.g., an ethics committee). Please also provide contact information for a data access committee, ethics committee, or other institutional body to which data requests may be sent.

Response: We have stated that data from this study are available upon request. This is due to Ethical issues (data contain sensitive patient information that cannot be disclosed and can be used for research only). The data can be found upon reasonable request from corresponding author (belayjanimen@gmail.com). 

Comment given: PLOS requires an ORCID iD for the corresponding author in Editorial Manager on papers submitted after December 6th, 2016. Please ensure that you have an ORCID iD and that it is validated in Editorial Manager. To do this, go to ‘Update my Information’ (in the upper left-hand corner of the main menu), and click on the Fetch/Validate link next to the ORCID field. This will take you to the ORCID site and allow you to create a new iD or authenticate a pre-existing iD in Editorial Manager.

Response: Comment accepted. 

Comment given: Your ethics statement should only appear in the Methods section of your manuscript. If your ethics statement is written in any section besides the Methods, please move it to the Methods section and delete it from any other section. Please ensure that your ethics statement is included in your manuscript, as the ethics statement entered into the online submission form will not be published alongside your manuscript.

Response: Comment accepted and ethics section moved to Methods section (see page 8 Ethics approval section)

Comment given: We note you have included a table to which you do not refer in the text of your manuscript. Please ensure that you refer to Table 5 in your text; if accepted, production will need this reference to link the reader to the Table.

Response: Table 5 is already mentioned on page 11 in line with Table 4 [Table 4 and 5] (on section “Factors associated with vancomycin resistant Enterococcus gut colonization”)

Comment given: Please include your tables as part of your main manuscript and remove the individual files. Please note that supplementary tables (should remain/ be uploaded) as separate "supporting information" files.

Response: Comment accepted and all Tables included in the manuscript (see page 8-12 for Table 1-5)

Comment given: Please include captions for your Supporting Information files at the end of your manuscript, and update any in-text citations to match accordingly.

Response: Comment accepted.

Comment given: The reviewers have commented on the manuscript and raised number of pertinent points that need to be addressed. Discussion section should interpret the results and discuss in the light of current knowledge rather than repeating the results. This applies to Conclusion section abstract too. Please revise the manuscript addressing all reviewer comments.

Response: Comment accepted and revisions were made according to comment give by reviewers and see the responses given below:-

2. Response to Reviewers 

Comment given: The language should be improved.

Response: We tried to amend and also re-write some parts of the manuscript. Check the manuscript again [please check for Abstract, Introduction, Materials and Methods, Result, Discussion, Conclusion and Limitations of the study sections.]

Comment given: The manuscript presents extensive introduction which can be reduced to two or three paragraphs.

Response: Comment accepted and Introduction section reduced to three paragraphs on the manuscript (see page 2)

Comment given: Statistical tests should be mentioned in the abstract also.

Response: Comment accepted and statistical tests added to Abstract section (“Bivariable and multivariable logistic regression model was used to analyze the association between dependent and independent variables. P-values in the multivariable analysis, adjusted odds ratio (AOR) and 95% confidence interval (CI) were used to determine the strength of association. P-value <0.05 was considered as significant.”)

Comment given: In methodology, some sentences can be eliminated like the section on antimicrobial susceptibility testing.

Response: Comment accepted and antimicrobial susceptibility testing deleted (page 4)

Comment given: The name of the institute should be masked.

Response: It is impossible to mask the name of the institute where research was conducted. If we mask the name of the institute where the research is conducted, the research will not be trustworthy to the scientific world.

Comment given: Discussion needs to be improved. Rather than stating the facts, the results should be compared and justification (or hypotheses) for any differences should be explored.

Response: Comment accepted and discussion section revised according to the comments.

Comment given: The disk diffusion test alone was used. Explain why MIC was not estimated and species level identification was not performed.

Response: Due to limitation of resources and budget, species level identification and MIC were not performed (see page 13 “Limitations of the study” section.)

---

## [Decision Letter · Decision Letter 1]

26 Apr 2021

PONE-D-20-34817R1

Vancomycin resistant enterococci and its associated factors among HIV infected patients on anti-Retro viral therapy in Ethiopia

PLOS ONE

Dear Dr. Regasa Dadi,

Thank you for submitting your manuscript to PLOS ONE. After careful consideration, we feel that it has merit but does not fully meet PLOS ONE’s publication criteria as it currently stands. Therefore, we invite you to submit a revised version of the manuscript that addresses the points raised during the review process.

Please address comments made directly on the manuscript. 

We look forward to receiving your revised manuscript.

Kind regards,

Iddya Karunasagar

Academic Editor

PLOS ONE

Journal Requirements:

Additional Editor Comments (if provided):

The reviewers have made some comments directly on the manuscript. Please address these

Reviewers' comments:

Reviewer's Responses to Questions

**Comments to the Author**

1. If the authors have adequately addressed your comments raised in a previous round of review and you feel that this manuscript is now acceptable for publication, you may indicate that here to bypass the “Comments to the Author” section, enter your conflict of interest statement in the “Confidential to Editor” section, and submit your "Accept" recommendation.

Reviewer #1: All comments have been addressed

2. Is the manuscript technically sound, and do the data support the conclusions?

Reviewer #1: Yes

3. Has the statistical analysis been performed appropriately and rigorously? 

Reviewer #1: Yes

4. Have the authors made all data underlying the findings in their manuscript fully available?

Reviewer #1: No

5. Is the manuscript presented in an intelligible fashion and written in standard English?

Reviewer #1: Yes

6. Review Comments to the Author

Reviewer #1: (No Response)

7. PLOS authors have the option to publish the peer review history of their article (what does this mean?). If published, this will include your full peer review and any attached files.

Reviewer #1: No

---

## [Author Response · Author response to Decision Letter 1]

28 Apr 2021

Point by point response

Comment given: Journal Requirements: Please review your reference list to ensure that it is complete and correct. If you have cited papers that have been retracted, please include the rationale for doing so in the manuscript text, or remove these references and replace them with relevant current references. Any changes to the reference list should be mentioned in the rebuttal letter that accompanies your revised manuscript. If you need to cite a retracted article, indicate the article’s retracted status in the References list and also include a citation and full reference for the retraction notice.

Response: Comment accepted and corrections were made on all references. (Check for tracking documents where changes were made on reference section, page 15-19)

Comment given: Table 3: G,P,TTC,MAC - all isolates are resistant to these groups. The table is conveying wrong information. .

Response: Comment accepted and correction made on the Table 3. All isolates are resistant to G, P, TTC, MAC 14 (100%); (G, P, TTC, MAC) + F 2 (14.28%) and (G, P, TTC, MAC) + F+ PH 11 (78.58%) (please check Table 3 page 9)

Comment given: It is unclear whether these are the studies conducted on HIV infected patients or not. If not, mention it. (Discussion section on page 11-12 paragraph 1)

Response: Comment accepted and correction made on the manuscript by declaring HIV status of studies done in different area (see page 11-12 paragraph 1 of Discussion section)

Comment given: No italics or upper case for “Enterococci” (page 13 paragraph 2).

Response: Comment accepted and re-written as Enterococci (please check on revised manuscript page 13 paragraph 2)

Comment given: correct as “were not done” on Limitation of study section which was written previously as:- “The isolated Enterococci were not identified to species level; molecular characterization and MIC were done due to resource limitation and budget constraints”

Response: Comment accepted and “were done” changed to “were not done” in limitation of the study section and re-written as “The isolated Enterococci were not identified to species level; molecular characterization and MIC were not done due to resource limitation and budget constraints” 

Comment given: Conclusion section paragraph 2 “Therefore periodic surveillance on antimicrobial resistance pattern of Enterococcus species is important for proper treatment, health professionals should strictly follow infection prevention protocols and further studies should be conducted on species identification and molecular characterization of Enterococci” marked as “Not a part of conclusion” by Reviewer

.

Response: Comment accepted and re-written as “Therefore, periodic surveillance on antimicrobial resistance pattern, adhering to rational use of antibiotics and implementing infection prevention protocols may reduce colonization by VRE” (see page 14 Conclusion section paragraph 2)

Comment given: Figure 1 not necessary as it can be expressed in text.

Response: Comment accepted and Figure 1 is expressed in words in Result section Incidence of vancomycin resistant Enterococcus sub section on page 8 

Comment given: Figure 2 change ug symbol

Response: Comment accepted and ug symbol is changed to µg.

Comment given: Have the authors made all data underlying the findings in their manuscript fully available? The PLOS Data policy requires authors to make all data underlying the findings described in their manuscript fully available without restriction, with rare exception (please refer to the Data Availability Statement in the manuscript PDF file). The data should be provided as part of the manuscript or its supporting information, or deposited to a public repository. For example, in addition to summary statistics, the data points behind means, medians and variance measures should be available. If there are restrictions on publicly sharing data—e.g. participant privacy or use of data from a third party—those must be specified.

Reviewer #1: No

Response: Comment accepted and Data availability statement added to page 14 that say “Data cannot be shared publicly because of ethical issues. However the data underlying the results presented in the study are available from corresponding author (belayjanimen@gmail.com) and Institutional Review Board (IRB) committee /Zerihun Zerdo (zedozerihun@gmail.com) on reasonable request.” (please check page 14 Data availability statement section)

Comment given: Remove tables from main text if presented at the end of the manuscript

Response: Tables are submitted in the main text only neither submitted individually nor the end of the manuscript

---

## [Decision Letter · Decision Letter 2]

3 May 2021

Vancomycin resistant enterococci and its associated factors among HIV infected patients on anti-Retro viral therapy in Ethiopia

PONE-D-20-34817R2

Dear Dr. Regasa Dadi,

We’re pleased to inform you that your manuscript has been judged scientifically suitable for publication and will be formally accepted for publication once it meets all outstanding technical requirements.

Kind regards,

Iddya Karunasagar

Academic Editor

PLOS ONE

Additional Editor Comments (optional):

All reviewer comments have been addressed.

Reviewers' comments:

Reviewer's Responses to Questions

**Comments to the Author**

1. If the authors have adequately addressed your comments raised in a previous round of review and you feel that this manuscript is now acceptable for publication, you may indicate that here to bypass the “Comments to the Author” section, enter your conflict of interest statement in the “Confidential to Editor” section, and submit your "Accept" recommendation.

Reviewer #1: (No Response)

2. Is the manuscript technically sound, and do the data support the conclusions?

Reviewer #1: (No Response)

3. Has the statistical analysis been performed appropriately and rigorously? 

Reviewer #1: (No Response)

4. Have the authors made all data underlying the findings in their manuscript fully available?

Reviewer #1: (No Response)

5. Is the manuscript presented in an intelligible fashion and written in standard English?

Reviewer #1: (No Response)

6. Review Comments to the Author

Reviewer #1: (No Response)

7. PLOS authors have the option to publish the peer review history of their article (what does this mean?). If published, this will include your full peer review and any attached files.

Reviewer #1: No

---

## [Editor Report · Acceptance letter]

21 May 2021

PONE-D-20-34817R2 

Vancomycin resistant *Enterococci* and its associated factors among HIV infected patients on anti-Retro viral therapy in Ethiopia 

Dear Dr. Regasa Dadi:

I'm pleased to inform you that your manuscript has been deemed suitable for publication in PLOS ONE. Congratulations! Your manuscript is now with our production department. 

Kind regards, 

on behalf of

Dr. Iddya Karunasagar 

Academic Editor

PLOS ONE